# Fantastic Rewards and How to Tame Them: A Case Study on Reward Learning for Task-Oriented Dialogue Systems

## Abstract

When learning task-oriented dialogue (TOD) agents, one can naturally utilize reinforcement learning (RL) techniques to train conversational strategies to achieve user-specific goals. Existing works on training TOD agents mainly focus on developing advanced RL algorithms, while the mechanical designs of reward functions are not well studied. This paper discusses how we can better learn and utilize reward functions for training TOD agents. Specifically, we propose two generalized objectives for reward function learning inspired by the classical learning to rank losses. Further, to address the high variance issue of policy gradient estimation using REINFORCE, we leverage the gumbel-softmax trick to better estimate the gradient for TOD policies, which significantly improves the training stability for policy learning. With the above techniques, we can outperform the state-of-the-art results on the end-to-end dialogue task on the Multiwoz 2.0 dataset.

## 1 Introduction

Task-Oriented Dialogue systems are designed to achieve a goal specified by a user in natural language. The classical approach to solve the task usually requires to solve several sub-tasks [1], including belief state tracking [2, 3], dialogue management (DM) [4], and natural language generation (NLG) for response generation [5]. More recently, end-to-end task-oriented dialogue based approaches [*e.g.*, 6–8] have been proposed, which significantly improve the overall performance. Besides, a number of works developed advanced reinforcement learning algorithms [*e.g.*, 9, 10] to further improve the over performance. However, the designs of reward functions are still heuristic based, which may lead to poor performance if they are not well tuned.

In this paper, we study how we can better learn and utilize reward function for training TOD agents. To be more concrete, we propose two generalized reward learning objectives in Section 3.1, and discuss how we can better utilize the reward function for dialogue agent training in Section 3.2 Further we empirically evaluate our proposed methods on Multiwoz 2.0 dataset in Section 4, which shows significantly improvements compared with previous state of the art approaches.

## 2 Background

**Task Oriented Dialogue as Reinforcement Learning.** We formulate the problem of task oriented dialogue systems as a partially observable Markov decision process (POMDP) [11], specified by $\mathcal{M} = \langle \mathbb{S}, \mathbb{A}, \mathbb{O}, \mathcal{P}, \mathcal{R}, \gamma \rangle$, where state $s \in \mathbb{S}$ consists of the previous dialogue history $h$ and the user intended goal $g$ specified prior to the start of the dialogue; $o \in \mathbb{O}$ is the observation that can be the user utterance; action $a \in \mathbb{A}$ can be the system response or dialogue act; $\mathcal{P}(s' \,|\, s, a)$ is the underlying transition probability; $\mathcal{R}(h, a, g)$ is the intermediate reward function for giving action $a$ under dialogue history $h$ and goal $g$; and $\gamma \in [0, 1]$ is the discount factor.

Submitted to 36th Conference on Neural Information Processing Systems (NeurIPS 2022). Do not distribute.

The dialogue history $h_t$ at timestep $t$ consists of all the previous observations and actions, *i.e.*, $h_t \triangleq \{o_0, a_0, \ldots, o_{t-1}, a_{t-1}, o_t\}$. Since the TOD agent can not directly observe the user goal $g$, it makes decision based on the entire dialogue history $h_t$ so far. Specifically, the policy $\pi$ is defined as a mapping from $h_t$ to a probability distribution over $\mathbb{A}$, *i.e.*, $\pi \triangleq \pi(a_t \,|\, h_t)$. The training objective is to find a policy $\pi$ that maximizes the expected (discounted) cumulative reward

$$ J(\pi) \triangleq \mathbb{E}_{\mu_g, \pi, \mathcal{P}} \left[ \sum_{t=0}^T \gamma^t \mathcal{R}(h_t, a_t, g) \right] , $$

where $\mu_g$ is the sampling distribution of goals and $T$ is the number of turns in a dialogue trajectory.

**Reward Design and Learning in Task Oriented Dialogue Systems.** Unlike classic RL problems where the intermediate reward function is well designed and provided, we can only get the evaluation metric at the end of the dialogue [12]. As a result, most of the existing works adopt the manually designed intermediate reward function that only gives binary reward to indicate success or not [*e.g.*, 13, 14, 10]:

$$ \mathcal{R}(h_t, a_t, g) := \begin{cases} R_{\text{const}}, & \text{if goal } g \text{ achieved at } t , \\ -R_{\text{const}} \text{ or } 0, & \text{if goal } g \text{ not achieved at } t , \end{cases} $$

where $R_{\text{const}}$ is a positive constant that can be 1. However, such sparse reward signals can be one of the reasons that the TOD agents learned by RL tend to have poor empirical performance [15].

To address the above issue, a few number of recent works focus on learning a dense reward function from demonstrations or mechanical dialogue assessments [*e.g.*, 16, 9], inspired by the reward learning from preferences in RL [17–19]. More precisely, suppose we are given two dialogue trajectories $\tau_i$ and $\tau_j$, with $\tau_i \triangleq \{g^{(i)}, (o_0^{(i)}, a_0^{(i)}), \ldots, (o_T^{(i)}, a_T^{(i)})\}$, and we want to learn a parameterized reward function $\mathcal{R}_\theta(o_t, a_t, g)$ with parameter $\theta$,[1] such that $\sum_{t=0}^T \mathcal{R}_\theta(o_t^{(i)}, a_t^{(i)}, g^{(i)}) > \sum_{t=0}^T \mathcal{R}_\theta(o_t^{(j)}, a_t^{(j)}, g^{(j)})$ when the preference score for $\tau_i$ is larger than $\tau_j$ ( denoted by $\tau_i \succ \tau_j$ for short). Then we can follow the Bradley-Terry model of preferences [20] to train the reward function by minimizing the following loss:

$$ \ell(\theta) = - \sum_{\tau_i \succ \tau_j} \log \left[ \frac{\exp\left( \sum_{t=0}^T \mathcal{R}_\theta(o_t^{(i)}, a_t^{(i)}, g^{(i)}) \right)}{\sum_{k \in \{i,j\}} \exp\left( \sum_{t=0}^T \mathcal{R}_\theta(o_t^{(k)}, a_t^{(k)}, g^{(k)}) \right)} \right] . \tag{1} $$

$\ell(\theta)$ can also be interpreted as a pairwise ranking loss, which is formalized as a binary classification in the problem of learning to rank [21–23].

# 3 Main Method

In this section, we start with proposed objectives for reward function learning based on classical approaches from learning to rank (LTR) literature [24], then we describe how to incorporate the learned reward function to training of MinTL to improve the overall performance.

## 3.1 Two Generalized Objectives for Reward Learning

We introduce two objectives `RewardNet` and `RewardMLE`, both of which can utilize multiple dialogue trajectories to optimize the reward function in a batch. Compared with the pairwise based approach described in Section 2, these two objectives can improve efficiency of the reward learning training, especially under the stochastic training settings.

**Setup.** Assume there are $N$ ($N \geq 2$) dialogue trajectories denoted by $\mathcal{D}_N \triangleq (\tau_1, \tau_2, \ldots, \tau_N)$, and each dialogue trajectory $\tau_i$ has an automatic evaluated metric score $S(\tau_i)$ (here we use combine score). For simplicity, we further assume the $N$ dialogue trajectories are ranked: $\tau_1 \succ \tau_2 \succ \ldots \succ \tau_N$, or equivalently $S(\tau_1) \geq S(\tau_2) \geq \ldots \geq S(\tau_N)$. Besides, we denote the accumulated reward of the dialogue trajectory $\tau_i$ by $J(\tau_i; \theta) := \sum_{t=0}^T \mathcal{R}_\theta(o_t^{(i)}, a_t^{(i)}, g^{(i)})$. And our goal is to learn the reward function $\mathcal{R}_\theta(o, a, g)$ such that the accumulated reward of the trajectories can reflect the ranking order: $J(\tau_1; \theta) \geq \ldots \geq J(\tau_N; \theta)$.

**RewardNet.** The proposed `RewardNet` objective for reward function learning is adopted from the *RewardNet* loss [25] in the LTR literature. Specifically, given the $N$ trajectories, we can define the `RewardNet` loss as the cross entropy between $\{J(\tau_i; \theta)\}_{i=1}^N$ and $\{S(\tau_i)\}_{i=1}^N$:

$$ \ell_{\texttt{RewardNet}}(\theta; \mathcal{D}_N) \triangleq - \sum_{i=1}^N P_S(\tau_i) \cdot \log \left( P_{J(\tau;\theta)}(\tau_i) \right) , \tag{2} $$

$$ \text{with} \quad P_S(\tau_i) = S(\tau_i) / \left( \sum_{k=1}^N S(\tau_k) \right), \quad P_{J(\tau;\theta)}(\tau_i) = \Phi(J(\tau_i; \theta)) / \left( \sum_{k=1}^N \Phi(J(\tau_k; \theta)) \right), $$

---

[1] We use the belief state, action and goal as the reward function input, and the belief state is part the observation $o_t$. We also drop the dependency on $h_t$ for $\mathcal{R}_\theta$ to simplify the reward function learning.

where $\Phi(\cdot)$ is a monotonic and positive function defined on $\mathbb{R}^+$, and $P_S(\tau_i)$ is the normalized prob. defined by the true score of each trajectory. Also, the pairwise loss proposed in CASPI [9] can be viewed as a special case of `RewardNet` loss where the number of trajectories $N = 2$.

**RewardMLE.** The `RewardMLE` objective is based on the *RewardMLE* loss [26], where we only utilize the ranking order in the batch dialogue trajectories $\mathcal{D}_N$, instead of the original metric scores $\{S(\tau_i)\}_{i=1}^N$. Let $y = \text{rank}(S)$ be the random variable that represents the rank order of the dialogue trajectories ($y(\tau_i) = i$ if the batch trajectories $\mathcal{D}_N$ are in rank order), then the `RewardMLE` objective is derived as the negative log-likelihood of the rank order $y$ under the Plackett-Luce choice model [27, 28] induced by $\{J(\tau_i; \theta)\}_{i=1}^N$:

$$\ell_{\texttt{RewardMLE}}(\theta; \mathcal{D}_N) :\triangleq -\log P\left(y \mid \{J(\tau_i; \theta)\}_{i=1}^N\right), \tag{3}$$

$$\text{with} \quad P\left(y \mid \{J(\tau_i; \theta)\}_{i=1}^N\right) = \prod_{i=1}^N \Phi(J(\tau_i; \theta)) / \left(\sum_{k=i}^N \Phi(J(\tau_k; \theta))\right),$$

where the trajectories in $\mathcal{D}_N$ are in ranked order as we described in the problem setup: $\tau_1 \succ \dots \tau_N$. In Eqs. (2) and (3), the monotonic function $\Phi$ transforms the unnormalized inputs $\{J(\tau_i; \theta)\}_{i=1}^N$ to a $N$-dimensional probabilistic simplex. We consider $\Phi$ as exponential function $\exp(\cdot)$ and power function $(\cdot)^p$ ($p \in \mathbb{N}$), which are also known as the softmax and escort transforms [29].

## 3.2 Policy Gradient Estimation with Learned Reward Function

With the learned reward function $\mathcal{R}_\theta(o, a, g)$, the next step is to improve the parametric dialogue agents $\pi_\phi$ via policy gradient [30]. Classical approach to estimate the policy gradient is via REIN-FORCE method [31]:

$$\nabla_\phi J_{\text{REINFORCE}}(\pi_\phi) = \mathbb{E}_\pi[\nabla_\phi \log \pi_\phi(a_t|h_t) G^\pi(h_t, a_t, g)], \tag{4}$$

where $G^\pi(h_t, a_t, g)$ is the discounted accumulated reward that the agents $\pi_\phi$ receives, starting from observation $o_t$ (part of $h_t$) and action $a_t$, given goal $g$. Previous work [9] indicates that when the discounted factor $\gamma > 0$, estimating $G^\pi(a_t, h_t, g)$ requires monte carlo sampling (on-policy) or temporal difference learning (off-policy), bot of which would require to learn an additional value function network. As a result, empirically we observe that it would introduce additional instability to the followed up end-to-end dialogue training. To simplify the training pipeline, we simply set the discounted factor $\gamma = 0$, and we know $G^\pi(h_t, a_t, g) = \mathcal{R}_\theta(o_t, a_t, g)$.

Though the policy gradient estimator defined in Eq. (4) is unbiased, it tends to have high variance, especially when the action space is large. As a result, the policy optimization with the REINFORCE estimator may diverge during the training. To address the high variance issue of REINFORCE estimator, we utilize gumbel-softmax trick [32, 33] to reduce the variance:

$$J_{\text{GS}}(\pi_\phi) = \mathbb{E}_{a_t \sim \pi(\cdot|h_t)}[\mathcal{R}_\theta(o_t, a_t, g)] = \mathbb{E}_{\boldsymbol{\epsilon} \sim \text{Gumbel}(0,1)}[R_\theta(o_t, f_\phi(h_t, \boldsymbol{\epsilon}), g)], \tag{5}$$

with

$$f_\phi(h_t, \boldsymbol{\epsilon}) = [f_\phi^{(1)}(h_t, \boldsymbol{\epsilon}), \dots, f_\phi^{(|\mathbb{A}|)}(h_t, \boldsymbol{\epsilon})] \in \mathbb{R}^{|\mathbb{A}|}, \quad \text{and} \quad f_\phi^{(i)}(h_t, \boldsymbol{\epsilon}) = \frac{\exp((\sigma_i(h_t; \phi) + \epsilon_i)/\lambda)}{\sum_{j=1}^{|\mathbb{A}|} \exp((\sigma_j(h_t; \phi) + \epsilon_j)/\lambda)},$$

where $\{\sigma_i(h_t; \phi)\}_{i=1}^{|\mathbb{A}|}$ are the logits of the categorical distribution defined by agent $\pi_\phi$. Note that $J_{\text{GS}}(\pi_\phi)$ is a *biased* gradient estimator for policy $\pi_\phi$. To achieve *bias-variance tradeoff*, we combine these two estimators to obtain the loss function for agent response generation:

$$\ell_{\text{GEN}}(\phi) := -(\alpha J_{\text{REINFORCE}}(\pi_\phi) + (1 - \alpha) J_{\text{GS}}(\pi_\phi)),$$

where $\alpha$ is a coefficient specified by users. Combining with the dialogue state tracking (DST) loss proposed in MinTL [6], we have the final loss for the end-to-end dialogue agent training:

$$\ell(\phi) = \ell_{\text{GEN}}(\phi) + \ell_{\text{DST}}(\phi). \tag{6}$$

## 4 Experiments

**Dataset.** We evaluate our proposed methods on the MultiWOZ 2.0 dataset [12], which is a representative TOD benchmark. MultiWOZ 2.0 is a large-scale and multi-domain dialogue corpus, consisting of conversations between a tourist (user) and a clerk (system) at an information center of a touristic city. This dataset has 8438 dialogues for the training set and 1000 dialogues for each of the validation and test set.

Table 1: Results of the end-to-end response generation task on the MultiWOZ 2.0 dataset. The best result on each metric is bold. The results of UBAR is from the reproduction by Jang et al. [10]. The results of CASPI is from our reproduction. All our provided results are the average over five random seeds.

| Algorithms | Inform | Success | BLEU | Combined Score |
|---|---|---|---|---|
| SFN + RL [34] | 73.80 | 53.60 | 16.90 | 83.10 |
| DAMD [35] | 76.40 | 64.35 | 17.96 | 88.34 |
| SimpleTOD [7] | 84.40 | 70.10 | 15.01 | 92.26 |
| MinTL [6] | 84.88 | 74.91 | 17.89 | 97.78 |
| SOLOIST [8] | 85.50 | 72.90 | 16.54 | 95.74 |
| UBAR [36] | 87.47 | 74.43 | 17.61 | 98.56 |
| GPT-Critic [10] | 90.07 | 76.63 | 17.83 | 101.13 |
| CASPI[9] | 91.37 | 82.80 | 17.70 | 104.78 |
| RewardNet: $N = 3$ (p=1) | 92.77 | 84.28 | 17.74 | 106.27 |
| RewardMLE: $N = 5$(softmax) | 91.49 | 83.38 | **18.97** | 106.40 |
| RewardNet:$N = 3$ (p=1) + GS | 92.63 | **84.32** | 18.35 | **106.83** |
| RewardMLE: $N = 5$ (softmax) + GS | **93.09** | 83.90 | 18.04 | 106.54 |

Table 2: Results on the simulated low resource settings, where 5%, 10%, and 20% of the training data is used to train the model. The best result on each metric under each setting is bold. "Comb." is the Combined Score. All our provided results are the average over five random seeds. Baseline results are from Lin et al. [6].

| Model | 5% | | | | 10% | | | | 20% | | | |
|---|---|---|---|---|---|---|---|---|---|---|---|---|
| | Inform | Success | BLEU | Comb. | Inform | Success | BLEU | Comb. | Inform | Success | BLEU | Comb. |
| DAMD | 56.60 | 24.50 | 10.60 | 51.15 | 62.00 | 39.40 | 14.50 | 65.20 | 68.30 | 42.90 | 11.80 | 67.40 |
| MinTL | 75.48 | 60.96 | 13.98 | 82.20 | 78.08 | 66.87 | **15.46** | 87.94 | 82.48 | 68.57 | 13.00 | 88.53 |
| RewardNet: $N = 3$ | 81.22 | 67.37 | 12.82 | 87.11 | **92.39** | **78.98** | 13.36 | **99.05** | 89.83 | **79.30** | 15.18 | 99.75 |
| RewardMLE: $N = 5$ | **82.90** | **69.61** | **14.26** | **90.51** | 89.67 | 77.48 | 14.80 | 98.38 | **90.15** | 78.70 | **15.81** | **100.24** |

**Evaluation Metrics.** Our proposed method is evaluated on the end-to-end dialogue modeling task of the MultiWOZ 2.0 dataset. Following the standard setup [*e.g.*, 12, 34], we use four automatic evaluations metrics: 1) **Inform** rate: the fraction of the dialogues where the system has provided an appropriate entity; 2) **Success** rate: the fraction of the dialogues where the system answered all the requested information; 3) **BLEU** score [37]: measures the fluency of the generated response; 4) **Combined Score** [34]: an overall quality measure defined as Combined Score $\triangleq$ (Inform + Success) $\times 0.5$ + BLEU. All our provided results are the average over five random seeds.

**Main evaluation.** Table 1 compares the performance of our methods with several classical and recent benchmarks, in the end-to-end response-generation task. As shown in Table 1, our proposed method not only improves the dialogue-task completion, measured by the Inform rate and the Success rate; but also generates fluent responses, reflected by the competitive BLEU scores. We note that the prior work CASPI is a special case of our proposed method when using the pairwise version of the RewardNet loss and when the probabilistic transform in Eq. (2) is the escort transform with power one. Comparing the result of CASPI with that of simply adding one more trajectory to estimate the RewardNet loss Eq. (2), we see that the RewardNet reward-learning loss improves the performance. As discussed in Section 3.1, our RewardNet approach considers more information for each update of the reward function, and thus could learn a more effective reward function.

We further improve the performance by changing the RewardNet loss Eq. (2) to the RewardMLE loss Eq. (3), with the softmax transform as in Xia et al. [26] and using two more trajectories to calculate the loss. This gain may come from the relative robustness of the RewardMLE loss to small errors in the scoring process, since the RewardMLE loss only uses the ranking of the provided scores, but not the numerical score values as in the RewardNet loss.

Adding policy-gradient updates via the Gumbel-softmax method improves the performance of both the RewardNet and RewardMLE models. This shows the efficacy of directly optimizing the response generation model *w.r.t.* the learned reward function.

**Low resource experiment.** We evaluate our models on the limited-data setting by following the testing strategy in Lin et al. [6]. Specifically, we use 5%, 10%, and 20% of the training data to train our models, RewardNet: $N = 3$ (p=1) and RewardMLE: $N = 5$ (softmax), and compare them with the baseline scores in Lin et al. [6]. Table 2 reports the results. It is clear that our models outperform the baselines, MinTL and DAMD, showing the efficacy of our proposed method. Comparing with Table 1, our models with 20% of the training data perform competitively with the baseline methods trained on the full training set.

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
