# OpenReview forum: "Fantastic Rewards and How to Tame Them: A Case Study on Reward Learning for Task-Oriented Dialogue Systems"
_NeurIPS.cc/2022/Workshop/LaReL — LaReL 2022_

### Official Review · Reviewer_mvok · 2022-10-14

**Rating:** 7
**Confidence:** 4

**Review:**

Summary: The paper tackles the problem of training language models with reward functions. It proposes and compares several methods for learning more dense reward functions that can give rewards per step, rather than giving simple binary rewards at the level of the entire trajectory. These learned reward functions are eventually optimized with a combination of REINFORCE and Gumbel softmax. Results appear to be SOTA on the MultiWOZ 2.0 dataset.

Significance:
This paper tackles the timely and useful problem of training language models with rewards, and appears to get SOTA results on an existing dataset. Thus, the paper is of significant interest to the LAREL community.

This work assumes that human raters can impose a total ordering over a series of trajectories in order to derive the rewards. Have the authors tested whether humans are willing and able to do this? How does this scale when a large sample size is needed?

Clarity:
The biggest weakness of the paper is its clarity, perhaps due in part to the compressed format. Some points that should be clarified:
- It is only possible to use the gumbel softmax formulation in this work because the learned reward function is differentiable. This would not be possible in another setting, such as if the reward depended on calling out to a game environment. This should be clarified in the text.
- Section 3.1 could be clarified. E.g. state earlier and more explicitly that the objective is to learn phi.
- It would be good to explain the significance of the MultiWOZ dataset earlier in the paper (i.e. abstract, intro), to make the significance of the work more clear.

Originality: Line 74 states that the RewardNet loss was developed in prior work. Is the main contribution then the RewardMLE loss? This should be clarified.

---

### Official Review · Reviewer_nXVf · 2022-10-19

**Rating:** 8
**Confidence:** 4

**Review:**

This papers investigates reward learning for task-oriented dialog learning. By learning a reward function from a pairwise ranking loss between two dialog trajectory and utilizing Gumbel-softmax to reduce variance, it achieves a new state-of-the-art on the MultiWOZ 2.0 dataset.

Strengths
- Easy to follow with clear formalization of the problem and method.
- Strong empirical results on a  common NLP benchmark for dialog response generation.
- Interesting ablation for low-resource setting.

Weaknesses
- The paper ends very abruptly. It is missing a conclusion & future work section.

Overall, this is a strong paper that should be of interest to LaReL attendees working on NLP with RL methods.

---

### Decision · Program_Chairs · 2022-10-21

Accept